# What Is the Contribution of Community Programs to the Physical Activity of Women? A Study Based on Public Open Spaces in Brazil

**DOI:** 10.3390/bs13090718

**Published:** 2023-08-29

**Authors:** Eduardo Irineu Bortoli Funez, Alice Tatiane da Silva, Letícia Pechnicki dos Santos, Ciro Romelio Rodriguez-Añez, Alexandre Augusto de Paula da Silva, Rogério César Fermino

**Affiliations:** 1Research Group on Environment, Physical Activity, and Health, Postgraduate Program in Physical Education, Federal University of Technology–Paraná, Curitiba 81310-900, Brazil; personal.funez@gmail.com (E.I.B.F.); ciroanez@utfpr.edu.br (C.R.R.-A.); 2Postgraduate Program in Physical Education, Federal University of Paraná, Curitiba 81531-980, Brazil; alicesilva@ufpr.br (A.T.d.S.); leticiapechnicki@ufpr.br (L.P.d.S.); 3Postgraduate Program in Health Sciences, Research Group on Physical Activity and Quality of Life, Pontifical Catholic University of Paraná, Curitiba 80215-901, Brazil; alexandre.augustosilva@outlook.com

**Keywords:** motor activity, exercise, parks, recreational, green areas, fitness centers, community participation, health promotion, public health, epidemiologic studies

## Abstract

Community programs can facilitate the access of vulnerable subgroups to physical activity (PA). This study analyzed the relationship between sociodemographic characteristics, health conditions, public open spaces (POS) usage, and women’s PA. The 155 participants were assiduous in taking part in PA classes in POS in São José dos Pinhais, Brazil. The accelerometer-based PA measures identified four outcomes: (1) daily light-intensity PA (LPA), (2) daily moderate-to-vigorous intensity PA (MVPA), (3) LPA in POS, and (4) MVPA in POS. Linear regression, and the Durbin–Watson and Mann–Whitney U tests were used for data analysis in STATA software. The main results showed that the weekly frequency (β: 10.9, *p* < 0.01) and intensity of the main activity in the POS (β: 22.4, *p* < 0.05) were related to daily MVPA. Economic level and length of stay in the POS were positively related to the LPA performed (*p* < 0.05). Weekly frequency (β: 2.4, *p* < 0.01), length of stay (β: 11.0, *p* < 0.01), and intensity of PA practiced in the POS (β: 5.9, *p* < 0.05) showed a positive relationship with MVPA in the POS. In conclusion, there was a positive relationship between some analyzed variables and PA of different intensities, especially the consistent relationship between weekly frequency of POS usage and MVPA. Participation in structured PA classes in a community program can contribute to an increase of 47 min of daily MVPA.

## 1. Introduction

Consistent evidence reports the benefits of regular physical activity (PA) in several health indicators in adults [1,2]. The World Health Organization (WHO) highlighted PA’s benefits, recommendations, and importance, even those of light or moderate-to-vigorous intensity (LPA and MVPA, respectively), for a short time daily [2]. On the other hand, insufficient PA is a significant global health problem, and most of the population is exposed to a high risk of chronic diseases [3,4]. About 28% of adults worldwide do not reach the WHO guidelines on PA, and the prevalence is even higher in Latin American countries (39%), especially in Brazil (47%) [4]. If there is no reduction in physical inactivity worldwide, it is estimated that around 500 million new cases of chronic diseases could occur by 2030 [4]. Of these, 74% of cases will occur in low-and middle-income countries. In addition to the burden on countries’ public health systems, the economic impact could reach USD 300 billion [4].

Physically active behavior is complex and determined by questions at various levels: individual (biological and psychological), interpersonal (social support and cultural norms and practices), environment (social, built, and natural environment), regional or national policy (e.g., urban planning and national PA plans), and global (e.g., economic development, and social and cultural norms) [5,6]. Many factors can affect choices and opportunities for being physically active, demonstrating that behavior is influenced by factors beyond motivation or the population’s knowledge of its health benefits [5,6].

In Latin American and low-middle-income countries, PA is lower in females, middle-aged and older adults, and those with low economic status and access to public open spaces (POS) for PA [5,7]. Thus, public policies should prioritize people with fewer possibilities and opportunities for access to PA. In this context, the community’s POS offers an important opportunity for social and leisure activities [8,9]. Also, evidence shows the additional benefits of PA in POS on mental and cardiometabolic health in adults [8,10]. Despite the recent interest in promoting POS usage for PA and leisure, it is essential to identify issues related to access to these places, particularly among vulnerable groups, including women [11]. The literature demonstrates that concern for personal safety is essential in restricting women′s access to PA and POS usage [11,12,13]. In this sense, the community programs developed in POS could help to reduce this cultural and social inequality of access to PA among vulnerable groups [14,15].

Different community interventions in POS, with PA promotion programs, have been developed worldwide, showing promising results [15,16,17]. In Latin America, community programs offer free PA classes to the population held in POS, such as parks, squares, and community centers [15,16]. In studies conducted in Brazil, for example, some of the correlates of participation in these programs include female gender, lower education, morbidities, neighborhood safety, and higher leisure-time PA (LTPA) [18]. Also, POS availability, proximity, access, quality, and offered activity can stimulate interest, making them more attractive and facilitating their usage for leisure activities, including PA and sports in the community [5,9,17,19,20,21,22]. In Colombia, the aerobics classes available in POS are more attractive and encourage high PA levels among women [23,24].

In the Brazilian context, POS usage is high (61%); and is positively associated with age group, quality of life, leisure opportunities, satisfaction with the place, and LTPA [25,26]. The weekly usage of parks can increase the probability of women walking in their leisure-time by five [25]. Nevertheless, these results are limited to parks or pocket parks in large cities, which limits the extrapolation of findings to community centers or smaller cities due to the differences in the availability, type, and quality of POS [9,12,17,26,27]. For example, the usage frequency of “sports and leisure centers” (a specific type of POS) is low (16%); however, their use increased walking during leisure time [21] by two. 

Limited evidence has analyzed the relationship between exposure to POS (e.g., availability, usage, and frequency of visits to POS) and accelerometer-based PA measures [25,28,29,30,31,32]. Schipperijn et al. [31] identified a positive relationship between proximity to POS in the neighborhood and MVPA, using pooled data from adults from eight countries. In the United States, Marquet et al. [30] found a positive relationship between green-area exposure and MVPA in older women, and Evenson et al. [28] observed a significant increase of 10 min of moderate-intensity PA (MPA) and MVPA during park usage days. In Barcelona (Spain), Vich et al. [29] identified that POS visits can increase significantly the PA of older adults by 28 min/day. Finally, in Curitiba (Brazil), Gonçalves et al. [32] found a negative relationship between the perceived proximity of walking facilities and LPA, and the absence of a relationship with MVPA in adults.

In 2017, the city hall of São José dos Pinhais (Brazil) implemented the “Active City, Healthy City Program”, which offers free PA classes in POS [21,33]. Among the users, 52% visit these places two or more times a week, and 45% participate in PA classes, such as aerobics, Zumba, and functional exercises [21]. Also, based on the observational method, women were 57% more likely to practice MVPA in these places [34]. Exploring PA correlates with different types of POS and how they can contribute to PA promotion in the population is essential to fill some knowledge gaps, especially in low-income and middle-income countries. The results could help guide and develop strategies and support the actions of policymakers to promote access to PA in POS, especially among women; at the same time, this could increase PA levels in other population subgroups more exposed to physical inactivity (e.g., older adults, those at a low economic level, overweight people, and people with morbidities) [5,9,17,19,22].

Therefore, this study aimed to analyze the possible relationship between sociodemographic characteristics, health conditions, POS usage, and PA of different intensities in women participating in the Active City, Healthy City Program.

## 2. Materials and Methods

The data used in this study were from the project “Effectiveness of community programs for PA promoting and reducing sedentary behavior in a medium-sized city in Latin America” [33]. The project aimed to evaluate the actions of the Municipal Secretary of Sport and Leisure. In compliance with the international recommendation, the study report was developed based on items from the Strengthening the Reporting of Observational Studies in Epidemiology (STROBE) list [35].

### 2.1. Design, Locals’ Contextualization, and Ethical Aspects

This quantitative, observational, and cross-sectional study was conducted with women participating in regular PA classes in POS in São José dos Pinhais, Brazil. The city is in the metropolitan region of Curitiba (capital of the Paraná state), has an area of 946 km^2^ (21% urban), a population of 323,340 inhabitants, a population density of 354 inhabitants/km^2^, and a human development index of 0.758 (high). Additional information is available in other publications [21,33,34].

The study was approved by the National Commission for Ethics in Research (CONEP) of the National Health Council, with a Certificate of Presentation of Ethical Appreciation, under protocol number 02088818.0.0000.0020, and by the Research Ethics Committee of the Pontifical Catholic University of Paraná, under number 3.071.682. Participants were consulted and informed about voluntariness and agreed to participate in the research by signing an informed consent form, according to the CONEP recommendations.

### 2.2. Selection of Public Open Spaces (POS) for Physical Activity (PA)

For this study, the POS that offered PA classes to the population was selected, locally known as “sports and leisure centers” [21,33,34]. These places are open daily (8 a.m.–10 p.m.) with free access, and many actions of the “Active City, Healthy City Program” are conducted, such as PA classes and usage of PA structures (courts, fitness zones, and walking paths), and sports events [21,33,34]. These characteristics are different from those found in urban parks in Brazil. The city has 13 “sports and leisure centers” in the urban area, and all were intentionally selected. However, five POS were excluded due to being unsafe for data collection, such as having differences in physical structure, or because they were evaluated in the pilot study. Thus, the data presented in this study refer to eight POS.

### 2.3. Participants Selection

Only women were intentionally selected to participate in the study, as they represent most people attending PA classes. In the first approach, the POS coordinator was asked for the number of PA classes for adults and older adults (*n* = 35) and the list of those enrolled in the respective classes (*n* = 547). Afterward, a simple random drawing was conducted based on the enrollment list, considering only women who attended classes at least once a week for three months (*n* = 304, average of 38 per POS). Due to team limitations and data collection logistics, it was decided to evaluate 20 women per POS (*n* = 160). However, based on a pilot study, a surplus of 20% was calculated in case of refusal or invalid data. In case of refusal, the next woman on the list of eligible women was invited. Women with physical limitations to PA, those with a low ability to understand the questionnaire or who had difficulties completing the activity diary, and those unable to use the accelerometer (ACC) were excluded (*n* = 3).

### 2.4. Data Collection

The sociodemographic information, health conditions, and POS usage were collected using a structured questionnaire and applied face-to-face individually in a reserved place so that there were no external influences on the responses (house or POS). The PA data were collected with an ACC. Data collection took place between July and December 2019. The average time of questionnaire administration was 13 min (±3 min, range 10–15 min).

### 2.5. Outcome Variable: Physical Activity (PA)

PA was measured with ActiGraph wb-GT3X + ACC [36] (Pensacola, FL, USA). Information recording began the day (midnight) following the delivery of the devices. The ACC was placed on the right side of the hip by an elastic strap, and the participants were instructed to use the equipment after waking up and remove it before usual sleep time or for activities in a liquid environment (e.g., bath, pool) [37,38].

The participants spent at least eight days with the ACC. Complementarily, an activity form was used for participants to record information regarding ACC usage (the time they woke up, went to sleep, placement, or removal for more than 30 min) [38]. Also, the POS’s entry and exit date and time, or whether they performed PA elsewhere, were recorded. To be considered a valid measurement, the ACC had to be used at least three days a week (≥600 min/day) and one day on the weekend (≥480 min/day) [37,38]. Raw data were collected in counts per minute (counts/min) and subsequently classified according to the cutoff points of intensities suggested in the literature [37] (0–99 counts/min: sedentary activity; 100–2689 counts/min: LPA; 2690–6166 counts/min: MPA; 6167–9642 counts/min: vigorous PA [VPA]; ≥9643 counts/min: very VPA).

PA was analyzed in two different situations: (1) the daily time that the participants spent on LPA and MVPA, and (2) only the time spent on LPA and MVPA during the period in POS (i.e., only during the time frame between the arrival and the departure from the site, recorded in the activity form) [28].

### 2.6. Predictor Variables

Based on the literature review, the possible predictor variables included sociodemographic characteristics, health conditions, and POS usage [9,20,21,25,26,39]. All variables were measured by valid and standardized questions or procedures utilized in previous studies and described below. The questionnaire was structured into seven domains and tested in a pilot study.

#### 2.6.1. Sociodemographic Characteristics

Age was categorized into three age groups (18–39, 40–59, and ≥60 yrs/old), and socioeconomic status (SES) was assessed using a standardized questionnaire widely used in research on PA in Brazil, and participants were classified into seven categories (A, B1, B2, C1, C2, D, E) [40]. For analysis, this variable was grouped into “economic levels” (low: SES C + D + E and high: SES A + B). Schooling was grouped into three categories of education (complete elementary school, complete secondary school, complete higher education, or more).

#### 2.6.2. Health Conditions

Body mass index (BMI) was calculated using self-reported body mass and height data, and participants were classified into two categories according to the cutoff points for adults (underweight or normal weight: BMI ≤ 24.9 kg/m^2^; overweight: BMI ≥ 25 kg/m^2^) and older adults (underweight or normal weight: BMI ≤ 26.9 kg/m^2^; overweight: BMI ≥ 27 kg/m^2^) [41]. The “medical diagnosis” self-report of morbidities such as hypertension, hyperglycemia, hypercholesterolemia, and hypertriglyceridemia established the presence of morbidities [42]. Participants were classified into four categories according to the number of their comorbidities (“0”, “1”, “2”, and “≥3”).

#### 2.6.3. Public Open Spaces (POS) Usage

The usage was measured with similar study questions about POS usage (e.g., parks), and adapted to the local context [25,26,39]. For example, the frequency of POS usage was evaluated by the question: “How often (how many times a week) do you come to the POS?”. The response options were arranged on a scale with seven options: (1) once a month, (2) a few times a month, (3) once, (4) twice, (5) three, (6) four, and (7) five times a week or more. For analysis purposes, the weekly frequency was grouped into four categories: “1–2”; “3”; “4” and “5 times a week”). Due to the inclusion criteria, among the participants, there were no women who used the POS once a month or a few times a month.

The length of stay in the POS was assessed with the question: “On a usual day, when you come to the POS, how long do you stay here?”, with responses in minutes, which later were classified into three categories: ≤59 min/day, 60 min/day, and ≥61 min/day. Usage time was evaluated with the question: “How long have you been regularly attending the POS?”. Participants reported time in months, and the variable was categorized into three levels (<1 year, 1–2 years, >2 years). The period of the day was evaluated with the question: “On which period do you attend the POS?” Responses were assessed dichotomously for each period (morning, afternoon, and evening).

Finally, the main activity performed at the POS was evaluated by the question: “What is the main activity you perform at the POS?” [25,39]. The response options were arranged in a list, with 12 possible and most common activities (stretching, gymnastics, walking, sports on the court, running, activities/classes at the gym, walking with the dog, sitting down to read, skateboard/bicycle/rollerblades, exercising at the fitness zone, and taking/accompanying the children [21,25]. For analysis, the activities were grouped into two categories according to the intensities of the Metabolic Equivalent of Tasks (METs): LPA (≤2.9 METs) and MVPA (≥3 METs) [43].

### 2.7. Data Quality Control

Data quality control was assured through ten steps [44,45]. First, the four administrators received 16 h of theoretical and practical training on technical administration procedures (approaching participants, ACC usage, recording losses, refusals, applying questionnaires, coding forms, and monitoring of the activity diary) based on the instruction manual prepared by the core project team. Second, the administrators strictly followed all these procedures; they were blinded to the objectives and hypotheses of the investigation and were supervised by a field coordinator. Third, a pilot study was conducted on a random sample of 20 women participants (described in the criteria in the Section 2.3) in PA classes from two POS to test the data collection procedures and the comprehension of the questions translated from other studies and adapted to the local context. Fourth, these two POS were excluded from the principal data collection to avoid contamination of the women’s data (e.g., research knowledge, reactivity in PA classes, and daily ACC usage). Fifth, the ACC was calibrated on a vibrating platform and presented ≥97% similarity in the measurements. Sixth, device usage was controlled during the week by telephone contact every two days. Seventh, the field coordinator conducted questionnaire data entry in EpiData software (EpiData Association, Odense, Denmark), and the ACC data analysis was conducted in ActiLife software. Eighth, data cleaning was carried out using exploratory analysis in SPSS software (v. 26.0, IBM SPSS Statistics, Armonk, NY, USA) to identify possible typing errors in data entry for each variable, detect outliers, and verify all variable distributions. Ninth, each variable outlier was personally checked in the questionnaire and activity form, and ACC data were downloaded and manually corrected in the database. And finally, all analyses were performed twice, reviewed, and validated by two authors who were experts in the built environment, POS usage, and PA (A.A.P.S. and R.C.F.).

### 2.8. Data Analysis

Data were represented with descriptive statistics (frequency distribution, mean, standard deviation [S.D.], minimum (Min.), maximum (Max.), median, interquartile range (IQR), and absolute and relative variation [∆] of minutes in PA, by intensity. Initially, the hypothesis of multicollinearity was discarded using the variance inflation factor (VIF) test ≤ 1.4. The independence of the residuals was analyzed using the Durbin–Watson test. The linear regression was applied to verify the relationship between possible predictors, the four PA outcomes, and the regression coefficient (β) and standard error of estimate (S.E.). First, bivariate analyses were performed to create the raw model. Afterward, four models were elaborated with the forced entry method, considering the following predictors: (1) sociodemographic variables; (2) health conditions; (3) POS usage, and, finally, (4) all variables from previous models. The Mann–Whitney U test compared the PA by POS usage day. Data were analyzed using STATA software (v. 16, StataCorp, College Station, TX, USA), and the significance level was kept at 5%.

## 3. Results

### 3.1. Participants’ Description

One hundred ninety women met the inclusion criteria and were invited to participate in this study. Of these, 7% (*n* = 13) refused, and 12% (*n* = 22) were excluded due to lack of valid ACC data. Therefore, the final analytical sample was 155 women. The largest proportion of women was in the 40–59 age group (49.0%, average age of 52.2 ± 4.3), had a low economic status (67.7%), and completed junior high school (45.8%). Regarding health conditions, most were overweight (57.4%) and did not report hypertension (78.1%), hyperglycemia (95.5%), hypercholesterolemia (85.2%), hypertriglyceridemia (95.5%), or comorbidities (67.7%) (Table 1).

Most women used the POS thrice a week (30.3%) and stayed there for more than 60 min (36.8%) and had attended the location for over two years (56.8%) (Table 1). The intensity of the main activity performed in POS was MVPA (72.9%) (Table 1). The activities performed were gym classes (78%), PA classes for older adults (19%), Pilates, activities at the fitness zone, and walking (1% each—data not presented in the Table 1).

### 3.2. Description of Accelerometer (ACC) Usage

On average, ACC was used for 6.5 days (±2.8), with 2.5 days (±1.5) at the POS. The average time spent at the POS was 66.4 min (±19.5) (Table 2).

### 3.3. Description of Physical Activity (PA)

The descriptive statistic of PA’s daily time median (min/day) is shown in Table 3. The daily time average (min/day) of PA was LPA: 739 (±171), MPA: 92 (±50), VPA: 8 (±11), and MVPA: 100 (±55) (Figure 1).

On average, women spent 14.4 min (±15.2) per week and 1.9 min (±2.0) per day on very VPA. At the POS, the time spent on very VPA was 8.3 min (±10.4) per week and 3.0 min (±3.9) per day (Table 4).

### 3.4. Relationship between Sociodemographic Characteristics, Health Conditions, POS Usage, and Women′s PA

On average, the daily time spent on LPA and MVPA was, respectively, 696.5 min/day (±126.5) and 96.9 (±42.1) (constant values in the crude model in Table 5 and Table 6, respectively). Sociodemographic characteristics, health conditions, and POS usage did not significantly relate to daily LPA (*p* ≥ 0.262) (Table 5).

In the crude model (bivariate), the highest weekly frequency (β: 11.8; S.E.: 2.8; *p* < 0.001), length of stay (β: 8.7; S.E.: 4.1; *p* < 0.01), and intensity of the main PA in POS (β: 21.0; S.E.: 7.4; *p* < 0.001) showed a significant relationship with daily MVPA (Table 5). In other words, changing one unit of the weekly frequency (1–2 for 3 times, for example) increases MVPA time by 11.8 min daily. In the final multivariate model (model 4), weekly frequency (β: 10.9; S.E.: 2.9; *p* < 0.01) and the intensity of the main PA practiced in the POS (β: 22.4; S.E.: 9.6; *p* < 0.05) maintained a significant relationship with daily MVPA. This model explains 14% of the variation in daily time spent on MVPA (*p* = 0.001) (Table 6).

On average, the time spent on LPA and MVPA during the time in POS was, respectively, 38.4 min/day (±15.9) and 28.6 (±14.6) (constant values in the crude model in Table 7 and Table 8, respectively).

Regarding the time spent on LPA in the POS, the final multivariate model (model 4) shows that the economic level (β: 5.9; S.E.: 2.7; *p* < 0.05) and the length of stay (β: 6.0; S.E.: 1.5; *p* < 0.01) showed a significant positive relationship with the outcome. Education level (β: −4.7; SE: 2.0; *p* < 0.05) and weekly frequency (β: −2.5; S.E.: 1.1; *p* < 0.05) showed a significant negative relationship. This model explains 21% of the variation in time spent on LPA during the time in the POS (*p* = 0.001) (Table 7).

Finally, in model 4, the weekly frequency (β: 2.4; SE: 0.8; *p* < 0.01), the time spent at the POS (β: 11.0; S.E.: 1.1; *p* < 0.01), and the intensity of the main PA (β: 5.9; S.E.: 2.6; *p* < 0.05) showed a significant positive relationship with the MVPA during the time in the POS. However, the economic level (β: −3.8; S.E.: 2.0; *p* < 0.05) was negatively related to this outcome. Model 4 explains 46% of the variation in MVPA in the POS (*p* = 0.001) (Table 8).

The descriptive statistic of the daily time (min/day) of PA by day of POS usage is shown in Table 9. The POS usage increases significantly (*p* < 0.001): the MPA (33 min/day, 41%), VPA (13 min/day, 325%), and MPVA (47 min/day, 56%) (Figure 2).

## 4. Discussion

This study analyzed the possible relationships between sociodemographic characteristics, health conditions, POS usage, and PA of different intensities of women participating in the Active City, Healthy City Program. The main results showed that the weekly frequency and intensity of the main activity in the POS were related to daily MVPA. Economic level and length of stay in the POS were positively related to the LPA performed. Weekly frequency, length of stay, and intensity of PA practiced in the POS showed a positive relationship with the MVPA in the POS. Participation in structured PA classes in a community program can contribute to an increase of 47 min of daily MVPA.

To the best of our knowledge, this is the first study in a medium-sized city in Latin America that analyzed the relationship between sociodemographic characteristics, health conditions, POS usage, and women’s PA. The methodological approach allowed exploration of the main individual predictors and four outcomes in PA, measured directly by ACC, in women participating in PA classes in eight different POS in the city. The sample represents 51% of those that attended PA classes at least once a week for at least three months at the selected POS. Then, the results can be extrapolated to persons with similar characteristics. The fact that the study explores the possible relationships between four different exposures to POS (weekly frequency, length of stay, time of usage, and intensity of main PA) and PA in women of different age groups (young, middle-aged, elderly), randomly selected, is also an essential aspect of the study that should be considered in interventions. Also, we minimize the risk of biases with high methodological quality control in data collection. According to the literature review, we did not find studies that have explored the contribution of community programs to PA of women based on structured and regular classes in POS, these being the strong and innovative points of the study.

The main results showed that weekly frequency, length of stay, and activity intensity performed at the POS were positively related to MVPA. These results reinforce the importance of POS in the neighborhood, with infrastructures, programs, and organized and structured classes to promote PA in women, regardless of age, education, economic level, and health conditions [15,16,17,18]. Identifying these characteristics can help public managers and coordinators of POS in neighborhoods to make decisions and direct resources and actions that promote PA at the community level.

Previous research has explored the association between other POS usage (parks and pocket parks) and LTPA. However, the difference in the structure of the POS and the self-reported method used to measure PA partially limit the extrapolation of results to understanding the importance of the different POS to the increase in daily PA and other intensities [21,25]. For example, the weekly frequency of parks was positively associated with an increase in the probability of walking ≥10 min/week and ≥150 min/week of about three and five times, respectively, in women during their leisure time [25]. Likewise, the analysis adjusted for gender showed that the higher weekly frequency in POS increased the probability of practicing walking and MVPA during leisure time by about two and three times, respectively [21].

The daily LPA was unrelated to individual, health, and POS-use predictors. These results were like those of Evenson et al. [28], who found no difference in the average minutes spent on LPA according to park visitations. In the present study, the absence of the relationship can be attributed to the participants’ homogeneity (age group, economic level, attending PA classes), which may have resulted in low variability in LPA. Despite the low variability of daily LPA, this information should not be interpreted negatively since the current WHO recommendations reinforce the importance and effect of any PA on health outcomes, especially lighter activities that are part of the daily life of a large population [2]. Also, women from low economic groups and low-and middle-income countries are more vulnerable to being physically inactive [22].

On the other hand, the weekly frequency and intensity of the main activity performed in the POS were positively related to daily MVPA. Similar studies are necessary for a proper comparison and discussion of these results. However, it is essential to highlight the contribution of these two factors to daily MVPA. Directing actions to encourage other forms of PA, such as active commuting to POS [2], is necessary. Yet, providing a safe urban environment for residents (e.g., tackling crimes and traffic safety) could increase the number of participants and adherence in these POS, in addition to the adequate guidance of the professionals responsible for PA classes [5,17,22,32]. Evidence also suggests that PA promotion programs in POS should be distributed according to cities′ social and spatial characteristics, aiming at reducing the disparity in access to services to reach the most vulnerable groups, including women [14].

Our results showed a significant difference of 47 min of MVPA (56%) during POS usage days. Similarly, Evenson et al. [28] found a significant increase of about 10 min/day (56%) on park visitation days. The greater magnitude of the difference observed in our study can be explained by the greater exposure to POS and the context of high-intensity, organized PA with professional guidance [23,24]. Also, the high-intensity PA observed while in the POS could be explained by the social support received during the classes [46,47]. Social support is a complex and multidimensional construct, conceptually understood as help or assistance from other people through relationships and social interactions with neighbors, family, spouses, health professionals, or those who organize PA classes [46]. Behavioral theories help us understand the possible influence or association between social support and PA [46]. Also, consistent evidence indicates that social support from friends/neighbors can positively impact the frequency, intensity, time, or type of PA (F.I.T.T) [5,46]. This influence is considerable when social support involves other people’s participation in activities, especially among women [46]. Indeed, studies suggest that community-based PA interventions improve cohesion and social capital among participants, increasing social support and PA [46,47].

The characteristics of the community and the program also help explain the relationship verified between POS usage and MVPA. For example, managers expect around 40% of the population to participate in an Active City, Healthy City Program action or activity [33]. However, what is observed in practice is that the same few people participate in some of the activities offered, mainly on weekends, in addition to PA classes and POS usage (participating and watching street running, events and activities sports, other PA classes or events, cultural and leisure activities, among others) [33]. Also, the low publicity of the program favors the vicious and virtuous cycle between knowledge, participation, and PA, observed by POS managers. These characteristics reinforce the hypothesis of a possible relationship between greater cohesion, capital and social support, and involvement in other PA in the community [9,12,46,47].

At least six limitations must be considered to interpret the results adequately. First, the sample was restricted to women who regularly participated in structured PA classes taught by professionals and in POS with areas for many activities. Thus, the data cannot be extrapolated to other contexts, such as for the population of women or men who do not regularly participate in PA or those who practice PA in private or paid places (e.g., gyms and health clubs). Second, the sample was relatively small, did not represent all people who practice PA in the other POS types in the city and was restricted only to women who regularly attend formal PA. Third, the measurement data from ACC for daily PA do not allow identifying in which of four domains the LPA and MVPA were performed (leisure, commuting, occupational, household). Fourth, the positive relationship between POS usage and PA could be explained by other factors that were not controlled in this study but are seen in similar studies (e.g., PA participation in other places or domains (for example, 76% of women reported walking or cycling to the POS)). Fifth, as with some self-reported data, there is a risk of self-report bias. Finally, as this was a cross-sectional study, we cannot conclude causality from our findings.

## 5. Conclusions

Weekly frequency was consistently related to three of the four PA outcomes. We observed a positive relationship between weekly frequency, intensity of activity, and daily MVPA. Economic and education level, weekly frequency, and time spent at the POS were related to LPA during time in the POS. Economic level, weekly frequency, time spent at the POS, and the intensity of the PA were related to the MVPA in the POS. The analyzed factors explain 46% of the variation in daily time spent on PA. Finally, participation in structured PA classes in a community program in POS can contribute to an increase of 47 min in daily MVPA.

These results may be helpful for public managers when making decisions regarding investments in programs of PA promotion, considering factors such as weekly frequency, intensity of activities, and length of stay in POS [9,17]. Future studies could verify the social, environmental, and political factors that favor women’s usage of POS. It is also suggested that a survey be carried out with a representative sample of residents living near these spaces to identify the prevalence, sociodemographic factors, health conditions associated with POS usage, PA based on the POS, and other correlates. These results can optimize the community’s access and usage of POS, especially among population groups with lower levels of PA, such as women and those from low economic groups.

## Figures and Tables

**Figure 1 behavsci-13-00718-f001:**
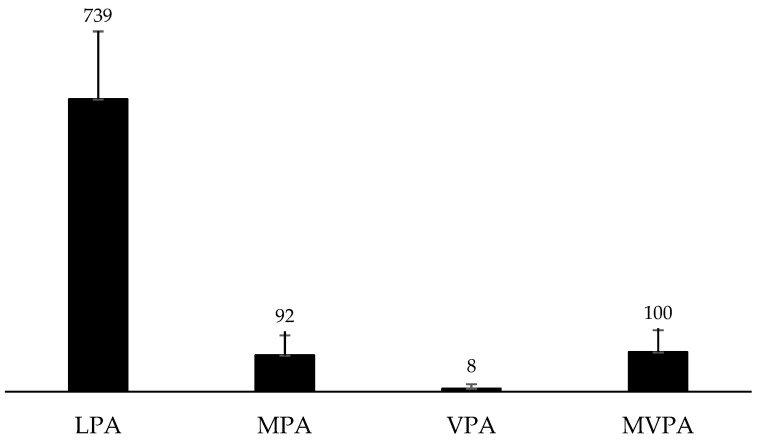
Average minutes of daily physical activity (PA) of women participating in the Active City, Healthy City Program, São José dos Pinhais, Brazil, 2019 (*n* = 155) (LPA: light-intensity PA, MPA: moderate PA, VPA: vigorous PA, and MVPA: moderate-to-vigorous intensity PA).

**Figure 2 behavsci-13-00718-f002:**
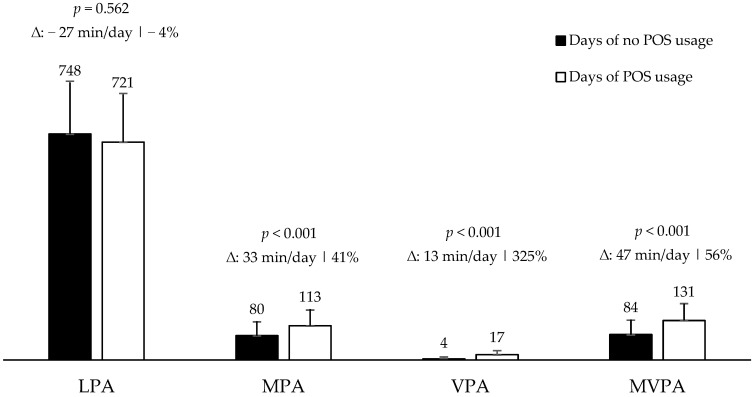
Average minutes of daily physical activity (PA) of women by POS usage day. São José dos Pinhais, Brazil, 2019 (*n* = 155) (LPA: light-intensity PA, MPA: moderate PA, VPA: vigorous PA, and MVPA: moderate-to-vigorous intensity PA).

**Table 1 behavsci-13-00718-t001:** Descriptive characteristics of the women participating in the Active City, Healthy City Program, São José dos Pinhais, Brazil, 2019 (*n* = 155).

Variables	Category	*n*	%
Sociodemographics characteristics			
Age group (yrs old)	18–39	50	32.3
	40–59	76	49.0
	≥60	29	18.7
Economic level	Low	105	67.7
	High	50	32.3
Education level	Elementary education or less	63	40.6
	Junior high school	71	45.8
	High school or more	21	13.5
Health conditions			
Body mass index (kg/m^2^)	Normal weight	66	42.6
	Overweight	89	57.4
Hypertension	No	121	78.1
	Yes	34	21.9
Hyperglycemia	No	148	95.5
	Yes	7	4.5
Hypercholesterolemia	No	132	85.2
	Yes	23	14.8
Hypertriglyceridemia	No	148	95.5
	Yes	7	4.5
Comorbidities *	0	105	67.7
	1	33	21.3
	2	15	9.7
	≥3	2	1.3
Public Open Space (POS) Usage			
Weekly frequency (times/week)	1–2	45	29.0
	3	47	30.3
	4	32	20.6
	5	31	20.0
Time spent at the POS (min/day)	≤59	44	28.4
	60	54	34.8
	≥61	57	36.8
Time of usage (yrs)	<1	26	16.8
	1–2	41	26.5
	>2	88	56.8
Intensity of main PA in POS	Light	42	27.1
	Moderate-to-vigorous	113	72.9

* Hypertension + hyperglycemia + hypercholesterolemia + hypertriglyceridemia, and PA: physical activity.

**Table 2 behavsci-13-00718-t002:** Description of an accelerometer usage by women participating in the Active City, Healthy City Program. São José dos Pinhais, Brazil, 2019 (*n* = 155).

	Average ± S.D.	Min.–Max.
Valid days	6.5 ± 2.8	4–13
Valid days in POS	2.5 ± 1.5	1–7
Usage time per week (min)	5807.2 ± 824.9	4413.0–9256.3
Usage time in POS per week (min)	189.4 ± 101.4	25.0–515.0
Usage time in POS per day (min)	66.4 ± 19.5	25.0–150.0

**Table 3 behavsci-13-00718-t003:** Minutes of daily physical activity (PA) of women participating in the Active City, Healthy City Program, São José dos Pinhais, Brazil, 2019 (*n* = 155).

Intensity	Median	Min.–Max.	IQR
LPA	737.1	366.7–1398.5	187.7
MPA	82.7	0.5–265.7	63.4
VPA	2.5	0.0–66.0	11.6
MVPA	90.4	0.5–271.9	72.8

LPA: light-intensity PA, MPA: moderate PA, VPA: vigorous PA, MVPA: moderate-to-vigorous intensity PA, and IQR: interquartile range.

**Table 4 behavsci-13-00718-t004:** Daily physical activity (PA) levels in public open spaces (POS) of women participating in the Active City, Healthy City Program, São José dos Pinhais, Brazil, 2019 (*n* = 155).

	Average ± S.D.	Min.–Max.
Daily PA level		
Time per week (min)		
Light	5116.7 ± 849.6	3467.8–8646.2
Moderate	639.6 ± 229.0	64.2–1246.0
Vigorous	45.8 ± 37.1	0.0–182.8
Very vigorous	14.4 ± 15.2	0.0–74.5
Time per day (min)		
Light	700.1 ± 138.7	462.4–1285.4
Moderate	91.2 ± 32.6	9.2–178.0
Vigorous	6.2 ± 5.0	0.0–26.1
Very vigorous	1.9 ± 2.0	0.0–10.6
PA level in POS		
Time per week (min)		
Light	105.6 ± 57.6	19.7–400.5
Moderate	58.5 ± 44.5	5.5–206.0
Vigorous	18.2 ± 20.5	0.0–102.5
Very vigorous	8.3 ± 10.4	0.0–49.5
Time per day (min)		
Light	38.9 ± 16.4	12.7–112.0
Moderate	20.2 ± 13.4	1.8–119.3
Vigorous	6.1 ± 5.4	0.0–25.7
Very vigorous	3.0 ± 3.9	0.0–29.5

**Table 5 behavsci-13-00718-t005:** Relationship between sociodemographic characteristics, health conditions, public open spaces (POS) usage, and women’s daily light physical activity (LPA). São José dos Pinhais, Brazil, 2019 (*n* = 155).

	Crude Model β (S.E.) min/day	Model 1 β (S.E.) min/day	Model 2 β (S.E.) min/day	Model 3 β (S.E.) min/day	Model 4 β (S.E.) min/day
*Constant*	696.5 ± 126.5	741.6 ± 49.4	716.4 ± 16.5	748.2 ± 49.0	812.0 ± 80.8
Sociodemographic characteristics					
Age group	−26.5 (14.3)	−26.8 (12.2)			−19.0 (21.2)
Economic level	8.4 (21.8)	6.2 (23.6)			7.5 (24.2)
Education level	8.2 (14.8)	−1.9 (16.9)			−7.3 (17.3)
Health conditions					
Body mass index	−27.2 (20.4)		−26.5 (20.5)		−25.3 (20.8)
Comorbidities #	−10.9 (13.1)		−10.2 (13.1)		−5.3 (14.2)
POS usage					
Weekly frequency	−3.3 (9.3)			−1.2 (9.3)	−1.2 (9.5)
Time spent	−20.7 (12.6)			−20.0 (12.7)	−41.6 (34.2)
Time of usage	−20.7 (13.3)			−16.4 (13.8)	−19.5 (13.2)
Intensity of main PA	23.9 (22.8)			17.2 (23.8)	−14.7 (14.2)
r^2^		2%	1%	2%	5%
*p*-value		0.332	0.309	0.262	0.503

Crude model: bivariate analysis; model 1: adjusted for sociodemographic characteristics; model 2: adjusted for health conditions; model 3: adjusted for variables of POS usage; model 4: adjusted for all variables used in models 1, 2, and 3; β: regression coefficient; S.E.: standard error of estimate; # number of morbidities; and r^2^: percentage value of model prediction magnitude.

**Table 6 behavsci-13-00718-t006:** Relationship between sociodemographic characteristics, health conditions, public open spaces (POS) usage, and women’s daily moderate-to-vigorous physical activity (MVPA). São José dos Pinhais, Brazil, 2019 (*n* = 155).

	Crude Model β (S.E.) min/day	Model 1 β (S.E.) min/day	Model 2 β (S.E.) min/day	Model 3 β (S.E.) min/day	Model 4 β (S.E.) min/day
*Constant*	96.9 ± 42.1	125.4 ± 16.2	103.4 ± 5.5	44.3 ± 15.2	62.0 ± 25.0
Sociodemographic characteristics					
Age group	−7.6 (4.8)	−10.8 (5.0) *			−0.9 (6.5)
Economic level	3.7 (7.2)	9.3 (7.7)			5.2 (7.4)
Education level	−6.2 (4.9)	−12.0 (5.5) *			−8.1 (5.3)
Health conditions					
Body mass index	−8.3 (6.8)		−8.1 (6.8)		−8.9 (6.4)
Comorbidities #	−4.2 (4.2)		−4.0 (4.3)		−2.8 (4.4)
POS usage					
Weekly frequency	11.8 (2.8) **			11.4 (2.9) **	10.9 (2.9) **
Length of stay	8.7 (4.1) *			6.1 (3.9)	5.0 (4.1)
Time of usage	2.0 (4.4)			1.5 (4.3)	1.4 (4.3)
Intensity of main PA	21.0 (7.4) **			22.0 (7.3) **	22.4 (9.6) *
r^2^	21.0 (7.4) **	9%	2%	14%	14%
*p*-value		0.052	0.312	<0.001	<0.001

Crude model: bivariate analysis; model 1: adjusted for sociodemographic characteristics; model 2: adjusted for health conditions; model 3: adjusted for variables of POS usage; model 4: adjusted for all variables used in models 1, 2, and 3; β: regression coefficient; S.E.: standard error of estimate; # number of morbidities; and r^2^: percentage value of model prediction magnitude; * *p* < 0.05; ** *p* < 0.01.

**Table 7 behavsci-13-00718-t007:** Relationship between sociodemographic characteristics, health conditions, public open spaces (POS) usage, and women’s light physical activity (LPA) during the time in the POS. São José dos Pinhais, Brazil, 2019 (*n* = 155).

	Crude Model β (S.E.) min/day	Model 1 β (S.E.) min/day	Model 2 β (S.E.) min/day	Model 3 β (S.E.) min/day	Model 4 β (S.E.) min/day
*Constant*	38.4 ± 15.9	29.4 ± 5.9	36.9 ± 2.0	38.3 ± 5.6	36.1 ± 9.1
Sociodemographic characteristics					
Age group	5.9 (1.7) **	4.7 (1.8) *			1.5 (2.4)
Economic level	3.0 (2.7) *	6.5 (2.8) *			5.9 (2.7) *
Education level	−4.8 (1.8) **	−5.1 (2.0) *			−4.7 (2.0) *
Health conditions					
Body Mass Index	3.3 (2.5)		3.4 (2.6)		3.5 (2.4)
Comorbidities #	−1.0 (1.6)		−1.0 (1.6)		−1.2 (1.6)
POS usage					
Weekly frequency	−1.4 (1.1)			−2.2 (1.0) *	−2.5 (1.1) *
Length of stay	6.2 (1.5) **			6.7 (0.3) **	6.0 (1.5) **
Time of usage	3.3 (1.6) *			1.6 (1.6)	1.0 (1.6)
Intensity of main PA	−8.6 (2.8) **			−8.3 (2.7) **	−6.3 (3.5)
r^2^		10%	1%	17%	21%
*p*-value		<0.001	0.318	<0.001	<0.001

Crude model: bivariate analysis; model 1: adjusted for sociodemographic characteristics; model 2: adjusted for health conditions; model 3: adjusted for variables of POS usage; model 4: adjusted for all variables used in models 1, 2, and 3; β: regression coefficient; S.E.: standard error of estimate; # number of morbidities; and r^2^: percentage value of model prediction magnitude; * *p* < 0.05; ** *p* < 0.01.

**Table 8 behavsci-13-00718-t008:** Relationship between sociodemographic characteristics, health conditions, public open spaces (POS) usage, and women’s moderate-to-vigorous physical activity (MVPA) during the time in the POS. São José dos Pinhais, Brazil, 2019 (*n* = 155).

	Crude Model β (S.E.) min/day	Model 1 β (S.E.) min/day	Model 2 β (S.E.) min/day	Model 3 β (S.E.) min/day	Model 4 β (S.E.) min/day
*Constant*	28.6 ± 14.6	36.6 ± 5.7	30.5 ± 10.0	9.2 ± 4.2	14.4 ± 6.8
Sociodemographic characteristics					
Age group	−3.3 (1.6) *	−3.4 (1.7) *			−2.1 (1.8)
Economic level	−1.0 (2.5)	−1.5 (2.7)			−3.8 (2.0) *
Education level	0.8 (1.5)	1.0 (1.9)			2.3 (1.4)
Health conditions					
Body Mass Index	−2.5 (2.3)		−2.5 (2.3)		−2.1 (1.7)
Comorbidities #	−1.0 (1.5)		−1.1 (1.5)		0.8 (1.2)
POS usage					
Weekly frequency	3.5 (1.0) **			2.5 (0.8) **	2.4 (0.8) **
Length of stay	11.0 (1.1) **			10.4 (1.0) **	11.0 (1.1) **
Time of usage	0.08 (1.7)			−1.3 (1.1)	−1.0 (1.2)
Intensity of main PA	9.0 (2.5) **			8.0 (2.0) **	5.9 (2.6) *
r^2^		1%	1%	45%	46%
*p*-value		0.235	0.456	<0.001	<0.001

Crude model: bivariate analysis; model 1: adjusted for sociodemographic characteristics; model 2: adjusted for health conditions; model 3: adjusted for variables of POS usage; model 4: adjusted for all variables used in models 1, 2, and 3; β: regression coefficient; S.E.: standard error of estimate; # number of morbidities; and r^2^: percentage value of model prediction magnitude; * *p* < 0.05; ** *p* < 0.01.

**Table 9 behavsci-13-00718-t009:** Minutes of daily physical activity (PA) by public open spaces (POS) usage day of women participating in the Active City, Healthy City Program, São José dos Pinhais, Brazil, 2019 (*n* = 155).

	Days of No POS Usage			Days of POS Usage			All Day		
Intensity	Median	Min.–Max.	IQR	Median	Min.–Max.	IQR	Median	Min.–Max.	IQR
LPA	738.3	366.7–1398.5	175.5	728.5	366.7–1334.7	215.0	737.1	366.7–1398.5	187.7
MPA	72.0	0.5–265.7	57.2	105.5 *	8.0–265.7	70.5	82.7	0.5–265.7	63.4
VPA	1.2	0.0–66.0	2.8	14.7 *	0–66.0	16.5	2.5	0–66.0	11.6
MVPA	76.2	0.5–271.9	59.5	123.2 *	9.2–271.7	75.2	90.4	0.5–271.9	72.8

LPA: light-intensity PA, MPA: moderate PA, VPA: vigorous PA, and MVPA: moderate-to-vigorous intensity PA, IQR: interquartile range, * *p* < 0.001.

## Data Availability

Data sharing is not applicable.

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
