# Peer review of "What Is the Contribution of Community Programs to the Physical Activity of Women? A Study Based on Public Open Spaces in Brazil"

_behavsci, 2023, doi:10.3390/bs13090718_

Round 1

Reviewer 1 Report

This manuscript is an original investigation and the topic under study is relevant.

From the point of view of methodological rigor, the problem of the study was identified. The objectives should be more clear, and more scientific evidence should be presented to support the problem under study. The research variables were presented. The procedures and instruments for the research should be better defined, explained and fundaments with bibliography references. The results, discussion and conclusion are presented.

Below are some suggestions and questions to consider.

In Materials and Methods chapter should be presented the bibliography reference that support the STROBE list, and should be explain, in short, what was the project “Effectiveness of 92 community programs for PA promoting and reducing sedentary behavior in a 93 medium-sized city in Latin America”, considering that the data used in this study were from that project.

All the criteria used to select the POS for PA should be more clear. It is suggest to specify the criteria instead of refer “among others” (line 118).

Should be indicated the age mean and standard deviation, of women selected to participat in the study.

Concerning Data Collection it was refere that “some data were collected through face-to-face interviews on the delivery day 139 of the accelerometer (ACC) in an individual, reserved place to have no external 140 influence on the responses (house or POS)”, and the other data, how it was?

Age was categorized into three age groups (18-39, 40-59, and equal or under 60 yrs/old). What was the bibliography used to fundament that groups?

SES was assessed using a standardized questionnaire, and the participants were classified into seven categories, should be refer what was the questionnaire used and the reference of its validation and publication.

To classified the participants health conditions and POS should be presented the literature fundaments, for understand the methodology presented.

Concerning the Data Quality Control should be present the fundaments and bibliography references to support the ten steps used.

In References Chapter should be included the articles DOI. Beyond the journals cited, It is also suggested to use other kinds of journals to fundament some options that you made, for example in material and methods chapter.

It is suggest to improve English language.

Author Response

We thank the two reviewers who evaluated the paper "What is the Contribution of Community Programs to the Physical Activity of Women? A Study Based on Public Open Spaces in Brazil" (ID behavsci-2481677), submitted on June 16, 2023. The comments and suggestions certainly contributed to the improvement of the manuscript. Below are the "point-to-point" responses to each reviewer's comments. Corrections and changes were made in the paper attached and are highlighted in yellow. We believe that all requests have been met or duly justified.

Reviewer 2 Report

See attachment.

The quality of English in this paper does need to be improved. Some comments in the feedback are centred on this.

Author Response

(The authors gave the same response as above.)

Round 2

Reviewer 1 Report

The authors made a new revision considering and based on the previous sugestions and comments. It was included new information that improve the study.

Author Response

We appreciate the reviewer’s consideration.

No action, correction, or response is required.

Reviewer 2 Report

Firstly, I thank the authors for taking the time to respond to the comments provided in the first round of review. I can see that some improvements have been made. Secondly, I still have some comments based on the paper and your responses provided. Please see them below.

Abstract

Line 22: It is still not clear what the four accelerometer-based outcomes are based on the wording. Perhaps consider changing to, “…: daily light-intensity PA (LPA) moderate-to-vigorous intensity PA (MVPA) outside of POS, and LPA and MVPA during POS.” Currently it reads like you are just repeating LPA, but I see, after a few reads, that it’s LPA and MPVA but they take place inside and outside of POS.

Line 27: Regarding your response, I understood at the time, it was just that it wasn’t clear in the abstract. It’s better now it has been modified.

Line 30: I do not refute that the results that you present here are important. I think you could do more to highlight why this particular study is novel and will create impact on the sample of focus. I still maintain that duration of the activities, weekly frequency and intensity of activities related to PA levels are not surprising. What you could do is highlight if this is new data for this particular demographic in your country, just as an example. Then it would make it more clear why this study is necessary (to understand better this demographic in this country so more individualised interventions can be developed specifically for them). It just needs more in the abstract to push the significance of this study.

Line 67: “offers”, not “offered”.

Methods

Line 149: “locally knowledge as…” does not read right, perhaps change to “…locally known as…” (if I’ve understood the meaning.

Line 177: Typically, research method-based terms like “interviews” are qualitatively aligned, and most researchers would interpret it this way. As per your response, you can run a quantitative study using questionnaires face to face, administered by researchers/administrators. So that your study can be properly understood, I would strongly suggest using words that align with the appropriate methodology. So, in this case, you could re-phrase to, “The average time of questionnaire administration was 13 minutes…”

Line 225: Good to see how the categories were numbered but it would be more helpful to also tell the reader what these numbers signify.

Line 227: similar study questions to what? I think I know what you mean but you should try to be as specific as possible so researchers could replicate your work, if necessary.

Line 251: possible missing end bracket

Line 256: similar to my comment above around appropriate methodological wording, I’d strongly advise changing “interviewer” to researchers or administrators. If you say interviewer, most readers will think an interview took place, when no interview took place; a series of quantitative questionnaires were administered.

Line 257: Similarly, “interview procedures” should be swapped to something like “administration procedures”, or something else, just not interview because an interview was not conducted.

Line 260: to emphasises my point, interviewers as a role, are not typically blinded because in qualitative methodology, the interviewer is part of the research method and analysis so they cannot and should not be blinded. I’m not saying you that you did a qualitative interview and that this should be changed, it’s good that the administrators were blinded, but it emphasises the point that the term “interviewer” should not be used.

Line 267: a word needs to be inserted after “women’s” because you shouldn’t then have “e.g.,” you should have a word which is then exemplified by your list of examples.

Tables 5-8: In your response you say that age is presented in tables 5-8, looking at those tables I see age as a sociodemographic characteristic; however it is not reported in the text. You would expect to find that age would impact different levels of PA and it’s interesting that it wasn’t significant for any of the levels, which in a way, is a finding in itself. If not reported here, I think it’s worth some consideration in the discussion, which I will look at in due course.

Discussion

Line 499: should be “methodological”.

Paragraph one: typically, the first paragraph in discussions should include the main aim of the study and the main findings. The start is fine but once the highlighted section starts it deviates from this. The highlighted information is warranted but it may not belong in this first paragraph. It also needs revisiting to check sentence structure and readability. The novelty and significance are worth noting here (first part of the paragraph) but strengths (last part of the paragraph) should belong in the strengths and limitations section.

Paragraph two: the highlighted sections here should be in paragraph one. The rest of the discussion should then be dedicated to discussing these main findings.

Line 522-525: this is a long sentence (67 words) and needs splitting, it also needs re-wording as either there are wrong words being used or missing words.

Line 533: should be “POS-use” to aid clarity in phrasing.

Line 534: need to stipulate what Evenson, said to remind the reader, if noted before, or to inform the reader so they know what you mean.

Line 576: where does the text in quotation marks come from? If from the study cited as 33, then page numbers are still required.

Line 580: same question as above.

Line 584: here is where your strengths would come in to join the limitations. With the limitations, suggestions for future research could be made.

Conclusion:

General comment: future work and practical implications should be provided before the conclusion where they have more space. They can then be referred to in the conclusion.

General comment: in your response, you say that your argument has been made in an earlier response. However, the novelty and significance of this work should still be emphasised in the conclusion to really drive home to the reader the importance of your work. I think some of the changes you have made to this section help you to do this but currently, because there are pieces that should be earlier in the discussion (see comment above), it dilutes the power behind the statements.

The quality of English language has improved since the last submission; however, further editing is required. Some points have been made in the comments provided; however, there were too many to include. The authors should take some more time to review the sentence structure and look out for missing words to ensure total clarity and accessibility to their work. 
